# Polycystic Ovary Syndrome: Challenges and Possible Solutions

**DOI:** 10.3390/jcm12041500

**Published:** 2023-02-14

**Authors:** Yue Che, Jie Yu, Yu-Shan Li, Yu-Chen Zhu, Tao Tao

**Affiliations:** Department of Endocrinology, Renji Hospital, Shanghai Jiao Tong University School of Medicine, Shanghai 200127, China

**Keywords:** polycystic ovary syndrome, pathogenesis, phenotype, treatment

## Abstract

Polycystic ovary syndrome (PCOS) is one of the most common endocrine disorders in women of reproductive age. This syndrome not only impairs female fertility but also increases the risk of obesity, diabetes, dyslipidemia, cardiovascular diseases, psychological diseases, and other health problems. Additionality, because of the high clinical heterogeneity, the current pathogenesis of PCOS is still unclear. There is still a large gap in precise diagnosis and individualized treatment. We summarize the present findings concerning the genetics, epigenetics, gut microbiota, corticolimbic brain responses, and metabolomics of the PCOS pathogenesis mechanism, highlight the remaining challenges in PCOS phenotyping and potential treatment approaches, and explain the vicious circle of intergenerational transmission of PCOS, which might provide more thoughts for better PCOS management in the future.

## 1. Introduction

Polycystic ovary syndrome (PCOS) is one of the most common endocrine disorders in women of reproductive age. It is characterized by oligo or anovulation, hyperandrogenism, and polycystic ovarian changes. This not only leads to female infertility but also affects the patient’s metabolic health. Furthermore, with the recognition of the “Developmental Origins of Health and Disease” (DOHaD) theory, the emphasis on the origin of adult diseases has been further shifted to gametogenesis and embryonic development. The metabolic disturbance of PCOS mothers before conception will increase the risk of PCOS in their offspring, leading to a “vicious cycle”. However, due to its strong clinical heterogeneity, the pathogenesis of PCOS is still unclear, which brings great challenges to clinicians concerning the correct diagnosis and treatment. What are the current trends in PCOS etiology exploration and remaining challenges in diagnosis or treatment? This paper reviews this issue.

## 2. Challenges and Possible Solutions of PCOS

### 2.1. Global Prevalence Trends of PCOS

The global prevalence of PCOS ranges from 6% to 21% [1], related to different diagnostic criteria, ethnicities, and regions. There were 1.55 million new instances of PCOS in women of reproductive age worldwide in 2017 [2], and 17.23% of these cases were women between the ages of 21–30 [3]. During the past 30 years, there have been significant increases in age-standardized incidences of PCOS in Asia. According to the 2003 Rotterdam criteria, the prevalence of PCOS in China was 10.01% in 2003, one of the countries with the highest increases in age-standardized incidence rates (73.53/100,000) [4].

PCOS is characterized by high risks of concurrent metabolic disturbances. Almost 50% of PCOS patients have obesity [5], 31.1% have impaired glucose regulation, and 7.5% have type 2 diabetes (T2DM) [6]. The relative risk of impaired glucose tolerance (IGT) in PCOS patients was 3.26 times higher, whereas the related risk of T2DM was 2.87 times higher than in healthy people [7]. Compared with non-obese PCOS patients, obese PCOS patients had a higher prevalence of metabolic syndrome (15.9% vs. 47.9%) and insulin resistance (7.1% vs. 27.8%) [8]. Under the subgroup analysis, Asian women with PCOS are more vulnerable to metabolic disturbances than other races, with a 5.2-fold increased risk of IGT and a 4.4-fold increased risk of T2DM compared with healthy women [7].

In conclusion, the global prevalence of PCOS has increased over the years. Its high risk of concomitant metabolic disorders may place a significant burden on the life-long health of PCOS women.

### 2.2. Deepening Understanding of the Etiological Mechanism of Polycystic Ovary Syndrome

In recent years, with the fast development in molecular genetics, high-throughput sequencing, transcriptomics, and proteomics, researchers have made great progress in genetics, epigenetics, gut microbiota, corticolimbic brain response, and metabolomics factors in PCOS pathogenesis (Figure 1).

#### 2.2.1. Genetic Variants

The biggest breakthrough in PCOS in the last decade has been genetics. A growing number of candidate gene studies have been performed. Analysis of Chinese Han genome-wide association data (GWAS) [9,10] showed 11 candidate loci for PCOS, including *THADA*, *LHCGR*, *DENND1A*, *C9ordf3*, *YAP1*, *RAB5B*, *HMGA2*, *TOX3*, *INSR*, *SUMO1P1*, and *FSHR*. The phenotype–genotype study showed that susceptibility variants in *THADA* and *INSR* conferred risk for metabolic syndrome, and variants of *DENND1A* and *TOX3* were associated with insulin resistance in PCOS women [9,10]. *INSR* and *TOX3* are significantly correlated to insulin resistance or metabolic syndrome [11]. Xia et al. [12,13] reported obesity and T2DM shared a common genetic basis to PCOS through serval loci (*ERBB3*, *FTO*, *PROX1*, *GIPR*, and *MC4R* for obesity [12], and *ADCY5*, *FTO*, *GIPR*, and *PPARG* for T2DM [13]). These findings provide rich information for the future prediction of reproductive outcomes and long-term complications in PCOS patients.

#### 2.2.2. DNA Methylation in Epigenetics

Although some PCOS patients show a high degree of familial aggregation, current GWAS analysis shows that the proportion of heritability accounted for by the PCOS loci is less than 10% [14]. This suggests that environmental and epigenetic mechanisms may play an important role in the etiology of PCOS. Epigenetic changes caused by adverse intrauterine or postnatal environments may trigger PCOS-like symptoms after birth, and such phenotypic changes are often inherited across three generations [15]. Lambertini et al. [16] used RNA-sequencing and genome-wide DNA methylation to analyze third-generation PCOS rats and found that ovarian DNA hypomethylation regulates key genes associated with inflammation, insulin signaling, and glucose metabolism. Furthermore, treatment of these third-generation prenatal Anti-Mullerian Hormone (AMH) exposed mice female offspring with methylating medication can reverse their PCOS-like neuroendocrine and metabolic alterations [16]. Sagvekar et al. [17] revealed that the alternation in transcriptional regulation of translocation methylcytosine dioxygenases (TETs) and DNA methyltransferases 3A (DNMT3A) may contribute to DNA methylation changes in cumulus granulosa cells of PCOS women. Nowadays, an increasing number of genome-wide association studies on prenatal hyperandrogenism-induced methylation in the ovarian tissue of PCOS rats are emerging, which will provide a more experimental basis for future PCOS treatment. This also has important implications for pregnancy counseling in PCOS women. Early identification of hyperandrogenism, treatment of their metabolism dysfunction to make them fit to have healthy offspring, and genetic diagnosis of the offspring for early prevention of possible subsequent metabolic problems will become a new area for the diagnosis and treatment of PCOS.

#### 2.2.3. Gut Microbiota Alternation and Brain-Gut Axis

The gut microbiota is particularly important for the impact of metabolic diseases. A previous study found that PCOS patients are more likely to have a high-sugar and high-fat diet [18], which can easily lead to microbiota disturbances. After the first finding of decreased α-diversity and β-diversity of PCOS patients [19], the relevance of PCOS pathogenesis to the gut microbiota was further investigated. A systemic review noted that the most common bacterial alterations in PCOS patients included Bacteroidaceae, Coprococcus, Bacteroides, Prevotella, Lactobacillus, Parabacteroides, Escherichia/Shigella, and Faecalibacterium prausnitzii [20]. Evidence showed that hyperandrogenism was negatively correlated to α diversity [21], while β diversity was reduced particularly in obese patients with PCOS [22]. The gut microbiota may alter the brain–gut axis, leading to appetite and energy metabolism dysfunction. Current research suggests that gut microbiota dysfunction disrupts the intestinal mucosal barrier, activates chronic inflammation, and produces various molecule metabolites that are involved in the development of PCOS [19]. However, the causal relationship between microbiota and PCOS is still unclear, and further studies are required to clarify it. In recent years, intestinal microbial preparations such as prebiotics have been increasingly used. A meta-analysis showed that prebiotics had the effect of reducing fasting insulin and triglycerides and increasing high-density lipoprotein (HDL) in PCOS patients, but these findings still lack a highly evidenced level of randomized controlled trial to identify [23]. Wang et al. [24] found that a 12-week high-fiber dietary intervention could reshape the gut microbiota of PCOS patients by enriching Bifidobacterium and Lactobacillus and effectively alleviate the PCOS clinical phenotypes. With the further development of gut microbiota sequencing technology and the maturation of germ-free mouse model technology, the gut microbiota may provide an effective way to carry out mechanism research and clinical treatment of PCOS in the future.

#### 2.2.4. Corticolimbic Brain Responses

As the various potential corticolimbic brain responses in the development of PCOS become better understood, exploring changes in the central nervous system in PCOS patients and their relationship with hyperandrogenism and hyperinsulinemia will provide more information for PCOS management in the future. Structural and functional brain imaging may play an important role [25]. Previous studies have shown that alterations in the white matter microstructure in PCOS patients are associated with eating patterns, mood disorders, cognitive dysfunction, and cerebral vascular disease [26,27,28]. While sex hormone levels have a broad impact on brain structure and activity, insulin resistance can affect the brain’s ability to respond to visual food cues [29]. However, these studies are limited to small samples. Future research advances may provide new clues to the possibility of new pharmacological targets for neuroendocrine dysfunction of PCOS patients.

#### 2.2.5. Metabolome Changes

Metabolites are small molecules that act as mediators and products of metabolism, which may provide new insight into many areas of disease. Serval studies have observed that PCOS symptoms are closely related to abnormal metabolites, such as glycerophospholipids [30], bile acids [31], branched-chain amino acids [32], and ceramides [33]. In addition to plasma samples, many metabolomics studies noted changes in the ovary and follicular fluid. Rice et al. [34] found that insulin-dependent lactate production was significantly impaired in granulosa-lutein cells from anovulatory PCOS women, whereas Sun et al. [35] demonstrated that there are significantly increased in free fatty acids, 3-hydroxynonanoyl carnitine, and eicosapentaenoic acid in the follicular fluid samples of PCOS patients who were undergoing in-vitro fertilization (IVF). In addition, Wang et al. [36] identified ganglioside GM3, ceramide, and pentacosatriene in fecal metagenomics as the predictivity of PCOS. It is hoped that advances in metabolomics knowledge will allow the identification of biomarkers to predict the future progression and complications of PCOS.

### 2.3. The Phenotype in PCOS

The most commonly used classification of PCOS is based on clinical manifestations. According to the Rotterdam criteria, the patient can be diagnosed with PCOS if any two of the three features are present: hyperandrogenism (HA), ovulatory dysfunction (OA), and polycystic ovarian morphologic features (PCO). There are four subtypes through the permutation of these three symptoms: OA + PCO, OA + HA, HA + PCO, and OA + HA + PCO. Patients with both OA and HA manifestations are so-called classic PCOS patients (OA + HA and OA + HA + PCO) [37]. Classic PCOS patients have the most severe problems, with more significant luteinizing hormone (LH) elevation, more severe hyperandrogenism manifestations, insulin resistance, dyslipidemia, a higher failure rate of ovulation induction, and a lower cumulative live birth rate after IVF and intracytoplasmic sperm injection (ICSI). In contrast, non-hyperandrogenic PCOS patients (OA + PCO) are the mildest. In some studies, the body mass index (BMI), homeostasis model assessment of insulin resistance (HOMA-IR), lipid metabolism, and other indicators of non-hyperandrogenic PCOS showed non-significant differences in healthy women.

In China, the classification of PCOS is more focused on the metabolic status of patients. Three phenotypes are classified according to other metabolism dysfunction: (1) the presence of obesity or central obesity; (2) the presence of impaired glucose tolerance, diabetes, or metabolic syndrome; (3) the presence of hyperandrogenism (classic PCOS as mentioned above). In addition, BMI-based classification is commonly used. Based on different diagnostic criteria for obesity, PCOS is classified as obese and non-obese PCOS. It is widely accepted that the pathological features of obese and non-obese PCOS are different. Non-obese PCOS patients tend to have more severe primary androgen metabolism dysfunction. Thus, a small amount of visceral fat accumulation can induce the occurrence of PCOS. Usually, such patients will have more pronounced clinical manifestations of hyperandrogenism, but the incidence of glucose and lipid metabolism disorders and cardiovascular diseases is relatively low compared to obese PCOS patients. On the other hand, the major abnormalities of androgen metabolism are relatively mild in PCOS patients with the obesity phenotype. Hyperandrogenism in obese PCOS patients is mainly caused by insulin resistance. Obese PCOS patients usually have more severe diabetes and hyperlipidemia, with a relatively higher risk of hypertension, fatty liver, and cardiovascular disease. The lean and obese phenotype is convenient for clinicians to differentiate. It is also a simple tool to predict the outcome of assisted reproduction and the risk of long-term metabolic complications.

Using biochemical and genotype data from the GWAS study of PCOS patients, Dapas et al. [38] investigated a new phenotypic subtype approach of PCOS. They revealed two distinct PCOS phenotypes: a “reproductive” group characterized by higher LH and sex hormone binding globulin (SHBG) levels with relatively low BMI and insulin levels, and a “metabolic” group characterized by higher BMI, glucose, and insulin levels with lower SHBG and LH levels. Furthermore, these subtypes were associated with novel and susceptibility PCOS candidate genetic loc. These findings fill the gap that the common-used phenotype approach of PCOS does not identify genetic subtypes and demonstrate that grouping all PCOS patients based on clinical presentation alone is insufficient to provide effective therapy in long-term outcomes.

### 2.4. The Advancement in PCOS Treatment

PCOS is a disease that affects women throughout their lifespans. As far as current management is concerned, the disease can only go into remission and not be cured. Therefore, different treatment priorities and goals should be set according to the different courses of PCOS, and this should never be limited to a single medication use. Future PCOS management should be more individualized and refined. Based on the guidelines and consensus, clinicians should pay attention to assessing the initiating factors of different phenotypes of PCOS (e.g., genetic background, environment, psychological stress, etc.) and manage them in an integrated biological–physical, and multidisciplinary way.

Lifestyle interventions are the first line of treatment for PCOS. First, PCOS patients should adhere to a good diet pattern. Based on an overall analysis of the patient’s previous dietary structure, clinicians should develop individualized dietary formulas, including macronutrient ratios, micronutrient intake, and total calorie restriction. Second, exercise is not only used as an aid to weight loss but also as a way to adjust the patient’s body structure. Based on different muscle-fat ratios, PCOS patients need an individualized exercise prescription that takes both fat loss and muscle gain into account. Finally, it is necessary to pay more attention to the psychological assessment of PCOS management.

In the prevention and treatment of metabolic-related comorbidities and complications of PCOS, it is crucial to advance the threshold of metabolic evaluation to the initial diagnosis rather than the patient having developed severe metabolic syndrome or cardiovascular diseases. In addition to lifestyle interventions, PCOS patients with metabolic dysfunction require other pharmacological treatments, such as metformin or statins (HMG-CoA reductase inhibitors).

However, not all PCOS patients respond to metformin. It was found that inositol combined with metformin is effective in improving insulin resistance and menstrual cycle frequency in infertile women with PCOS [39]. However, a recent meta-analysis showed that metformin did not improve the clinical pregnancy rates and live births of PCOS patients in IVF/ICSI [40]. For PCOS patients with hyperandrogenemia, there is evidence that metformin does not affect the reduction in androgen levels in pregnant PCOS women [41]. However, in a subgroup analysis, a modest androgen-lowering effect was observed in non-obese PCOS women with male fetuses [41]. It is unclear whether the androgen-lowering effects of metformin also occur in fetal circulation.

Therefore, other metabolically modifiable agents, including classic insulin sensitizers (thiazolidinediones) and non-classical insulin sensitivity-improving drugs (acarbose, sodium-glucose cotransporter (SGLT2) inhibitors, glucagon-like-peptide 1 receptor agonist (GLP-1RA)), are also indicated and effective in PCOS patients. Our study found that the combination of GLP-1RA and metformin improved the reversal of prediabetes in overweight/obese PCOS patients [42]. Other studies have also shown evidence that metabolic surgery plays an important role in improving metabolic dysfunction and hyperandrogenism in obese PCOS patients [43]. However, the effectiveness of metabolic surgery in preventing long-term metabolic complications and metabolic risk in subsequent generations remains unknown.

### 2.5. Treatment Dilemma in PCOS

As mentioned above, because of the highly heterogeneous nature of PCOS, the current clinical classification does not reflect the underlying pathological mechanisms. Individualized and precise treatment of PCOS is difficult due to the lack of a clear understanding of the etiology and prognostic markers. Furthermore, current national and international guidelines provide only experimental symptom management and lack mechanism-based treatment.

It cannot be ignored that the incidence of metabolic diseases in PCOS offspring is not encouraging. In addition to genetic and environmental factors, it has been suggested that the origin of metabolic diseases can be traced back to the early developmental stages of life. This theory was originally raised by David Barkers, emphasizing that maternal nutrient deficiencies during pregnancy would result in low-birth-weight infants having rapid catch-up growth [44]. These offspring will exhibit impaired insulin secretory capacity, glucose and lipid metabolism, and increased risk of metabolic syndrome in their early life. Today, this hypothesis enlarged the theory of Developmental Origins of Health and Disease (DOHaD) theory, which means that an adverse intrauterine environment may lead to irreversible lifelong consequences for offspring, increasing susceptibility to non-communicable diseases (e.g., obesity, T2DM, metabolic syndrome, cardiovascular disease) in later life [45].

Serval studies have proven that PCOS offspring may be at significantly higher risk for early onset metabolic disorders. Prenatal androgen-exposed offspring mice exhibit impaired glucose tolerance and increased visceral adiposity [46], elevated circulating triglyceride levels [47], left ventricular hypertrophy [48], liver lipogenesis imbalance [49], and significant accumulation of fat in the liver [50] in their offspring. According to Finland’s national birth cohort, maternal PCOS was significantly associated with a 1.58-fold increased risk of childhood obesity, a 1.37-fold increase in adolescent obesity, and a 2.06-fold increase in T2DM in early adulthood [51]. Zhang et al. prospectively tracked the growth trajectory of the Ningbo birth cohort and obtained similar results [52]. In addition, Risal et al. found that daughters of PCOS mothers had a five-fold increased risk of PCOS in a Swedish nationwide register-based cohort and a clinical case–control study from Chile [53]. The metabolic dysfunction in the PCOS offspring will further put grandchildren at risk, which in turn creates a vicious circle of intergenerational transmission (Figure 2). This will place a heavy burden on both the patient’s family and society. How to break this vicious cycle remains a huge challenge for the future.

According to guideline recommendations, lifestyle interventions and metformin are the preferred options for the treatment of PCOS. However, these classical treatments are not as effective in preventing the intergenerational transmission of PCOS and its associated metabolic dysfunction. Lifestyle interventions are proven effective in reducing the health risks of the offspring of patients with preconception PCOS in observational studies and preclinical animal models. Dhana et al. [54] found that mothers who live healthy lifestyles (quality diet, normal weight, regular exercise, light to moderate alcohol consumption, and no smoking) reduce their children’s risk of obesity by 75%. Xu et al. [55] found that offspring of high-fat diet-induced obese female mice had abnormal glucose tolerance and significant hepatic adipocyte degeneration at 12 weeks of age, while the mother switching from a high-fat to a standard diet 9 weeks before pregnancy can protect the offspring from metabolic dysfunction. However, there is still a lack of high-level evidence-based medical evidence from randomized controlled studies to demonstrate the effectiveness of lifestyle interventions in promoting the offspring’s health. Mintjens et al. [56] reported a 6-month preconception lifestyle intervention (including reduced energy intake, exercise, and health consulting) in 577 obese infertile women to reverse pre-pregnancy metabolic abnormalities and follow up their offspring until 4–6 years of age. However, no significant differences in metabolic markers such as height standard weight Z score, body fat percentage, blood pressure Z score, pulse wave velocity (PMV), blood lipids, glucose, blood insulin, and HOMA-IR were found in the offspring due to the limitations of high dropping rates and uncontrolled weight gain during pregnancy.

Metformin is the most commonly used medication in the treatment of PCOS [57]. Echiburu et al. [58] reported that metformin during pregnancy could protect the offspring from the DNA methylation levels change in leptin, lipocalin receptor 2, and androgen receptor genes caused by maternal PCOS. Meanwhile, it is safe to use metformin in the perinatal period. There is no evidence that perinatal metformin use will increase the risk of congenital malformations [59], miscarriage, and preterm delivery [60] or long-term effects on children’s average cognitive function [61]. However, Hanem et al. [62,63] reported the PedMet-study, which is the longest follow-up study on the offspring of PCOS patients who have taken metformin during pregnancy. The results showed that metformin use during pregnancy in PCOS patients might induce an increase in the adiposity of their offspring at 4 years of age, along with an increase in fasting glucose levels and systolic blood pressure at age 8 years of age. The increase in adiposity in childhood strongly suggests adult obesity and a substantially increased risk of metabolic syndrome and cardiovascular disease. This may suggest concerns about the effectiveness of metformin as a gestational intervention for PCOS mothers to reduce health risks in their offspring.

### 2.6. Future Prospective of PCOS Therapeutic Target

In recent years, various intervention targets have shown good therapeutic promise in preclinical models or clinical studies with small samples. There is evidence of brown fat dysfunction in PCOS patients and future potential to improve metabolic disorders in PCOS patients by activating brown fat [64]. Both PCOS mouse models and patients show whole-gene hypomethylation, and it has been shown that the clinical features of PCOS may have transgenerational transmission effects in mice by altering the DNA methylation status. S-adenosylmethionine (SAM), a medication that can reverse DNA methylation, may have a potential therapeutic effect on PCOS, which has been demonstrated in the preclinical models [65]. Animal studies have shown that supplementation with interleukin-22 (IL-22) or glycodeoxycholic acid (GDCA) ameliorates insulin resistance in B. vulgatus mice and significantly corrects estrous cycle dysfunction, alters ovarian morphology, and improves abnormal hormone levels [66]. Kisspeptin/Neurokinin B/Dynorphin (KNDγ) neurons are expected to be a new therapeutic target for modulating gonadotropin-releasing hormone (GnRH)/LH pulse generators in the future. A randomized, double-blind, placebo-controlled, multicenter (5 European clinical centers) Phase 2a study [67] evaluated the clinical efficacy and safety of the neurokinin 3 (NK3) receptor antagonist “fezolinetant” in the treatment of PCOS. The study found that fezolinetant has a sustained effect of suppressing hyperandrogenism and lowering the LH/ follicle-stimulating hormone (FSH) ratio. In addition, kisspeptin treatment increased oocyte production by more than 60% in IVF patients without increasing the risk of ovarian hyperstimulation and ovarian hyperstimulation syndrome (OHSS) [68]. Notably, although the results of preclinical studies and cohort studies suggest that kisspeptin improves reproductive hormone secretion and ovulation in some animal models of anovulation PCOS, the effect on PCOS patients remains unknown. Universally, future clinical studies could increase the number of patients and focus on the impact on their improved metabolic profile. In conclusion, based on the novel findings of PCOS etiology in genetic, epigenetic, and cortical brain responses, these effective therapeutic targets, such as SAM, IL-22 supplementation, and fezolinetant, will lead to a new area in the future management of PCOS.

At the same time, in addition to the exploration of new treatment targets, the timing of interventions for PCOS is also very important. Current evidence has shown that lifestyle interventions and metformin for PCOS during pregnancy are not effective in improving offspring health. In recent years, more and more people have begun paying attention to preconception health. Lifestyle improvement behaviors (e.g., good dietary structure, smoking and alcohol cessation, exercise, regular sleep, etc.) must take months or even years to achieve. It is advocated that preconception care should be extended to several years before conception to address preconception risk factors such as diet and obesity to achieve optimal fertility [69]. In the future, metabolic modulation in young women with PCOS, especially before their pregnancy plans, may be an effective solution to prevent the intergenerational transmission of PCOS and its metabolic disorders.

## 3. Conclusions

The global prevalence of PCOS has risen throughout the years. Its increased risk of concomitant metabolic illnesses may impose a major burden on PCOS women’s long-term health. Researchers have made significant advances in genetics, epigenetics, gut microbiota, and corticolimbic brain response variables in PCOS etiology. Metformin and lifestyle interventions have demonstrated encouraging results in the treatment of PCOS. Meanwhile, metabolically modifying drugs and metabolic surgery have an essential role in treating metabolic dysfunction and hyperandrogenism in obese PCOS patients. However, existing treatment guidelines only provide experimental symptom management instead of mechanism-based treatment. Meanwhile, the incidence of metabolic diseases in PCOS offspring is also concerning. Future PCOS management should pay more focus to novel therapeutic targets, and a sharper focus on intervention before conception is required to improve both maternal and child health in PCOS patients.

## Figures and Tables

**Figure 1 jcm-12-01500-f001:**
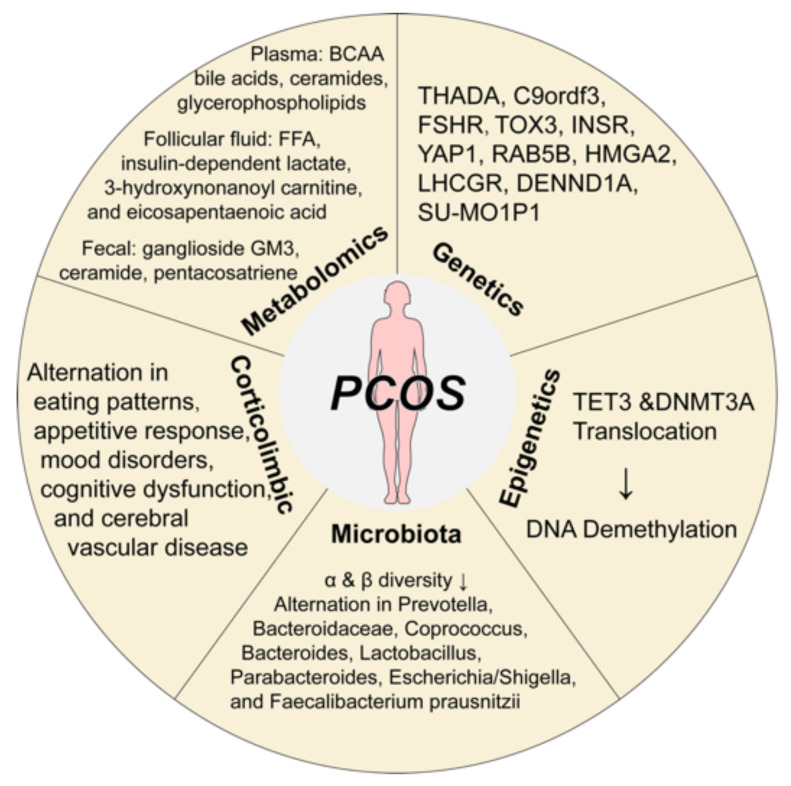
The pathological mechanisms of polycystic ovary syndrome. Great progress has been made in the etiological mechanism of PCOS, including genetics, epigenetics, microbiota, corticolimbic brain response, and metabolomics. Abbreviations: BCCA, branched-chain amino acid; DNA, Deoxyribonucleic acid; DNMT3A, DNA methyltransferases 3A; FFA, free fatty acid; FSHR, follicle-stimulating hormone receptor; INSR, insulin receptor; PCOS, polycystic ovary syndrome; TET, translocation methylcytosine dioxygenases. Components of this figure were created using Servier Medical Art, provided by Servier, licensed under a Creative Commons Attribution 3.0 unported license (https://smart.servier.com).

**Figure 2 jcm-12-01500-f002:**
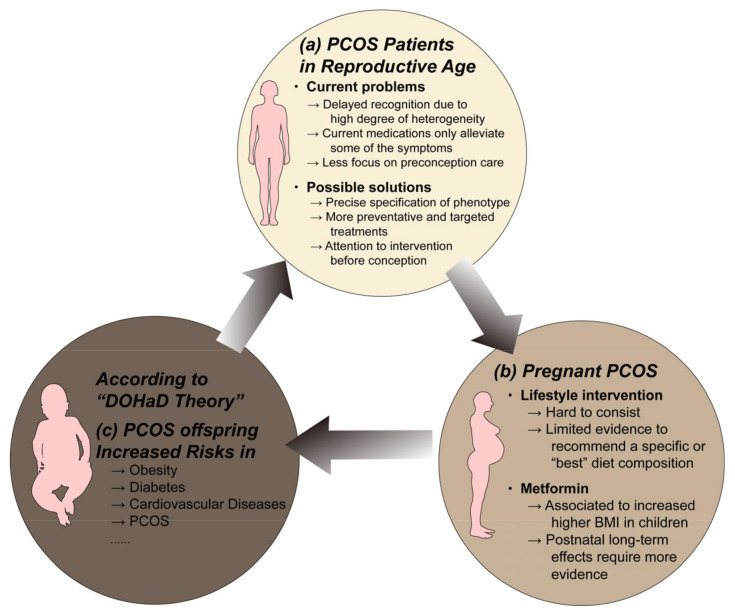
The vicious circle of intergenerational transmission of PCOS. (a) Many challenges remain in the diagnosis and treatment of PCOS in reproductive age. The concurrent metabolic disturbances in PCOS will bring risks to both mother's and offspring's health. (b) Lifestyle intervention and metformin are the first-line treatment for PCOS and they are safe for use during conception. However, their efficiency in benefiting PCOS offspring requires more evidence to demonstrate. (c) The adverse uterine environment caused by maternal PCOS, according to the DOHaD theory, may increase the susceptibility to non-communicable diseases of their offspring. The offspring of PCOS are vulnerable to serval metabolic diseases in their early adult life, which will also affect their children's health. How to break this vicious cycle remains a huge challenge for the future. Abbreviations: BMI, body mass index; DOHaD, developmental origins of health and disease; PCOS, polycystic ovary syndrome. Components of this figure were created using Servier Medical Art, provided by Servier, licensed under a Creative Commons Attribution 3.0 unported license (https://smart.servier.com).

## Data Availability

No new data were created or analyzed in this study. Data sharing is not applicable to this article.

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
