# Peer review of "Polycystic Ovary Syndrome: Challenges and Possible Solutions"

_jcm, 2023, doi:10.3390/jcm12041500_

Round 1

Reviewer 1 Report

This manuscript's title is misleading: "Are the authors referring to opportunities for research?"

A better title could be "Polycystic Ovary Syndrome: Challenges and Possible solutions"

The manuscript is not comprehensive and lacks coherence.

The introduction is very poorly written.

I find the language used in the manuscript to be unprofessional at several points.

This topic is interesting, but I don't think it will engage readers in a meaningful way.

In my opinion, the manuscript does not meet the standards of the journal in its current form. Extensive work will be needed to make the manuscript more comprehensive and meaningful. 

Author Response

Please see the attachment. Our point-by-point responses are listed in the attachment file. 

Reviewer 2 Report

The manuscript entitled "Polycystic Ovary Syndrome: Opportunities and Challenges" has been reviewed and the following issues need to be addressed to improve the manuscript:

1. The study is not fluid in language and needs extensive correction. Additionally, writing errors have been observed which have to be resolved ASAP.

2. The study has not included the latest findings: e.g., 10.1007/s43032-020-00430-0 

3. The manuscript needs tables and illustrations to enhance the representation of the text.

4. The text has claimed that the genetic etiology of PCOS has been discovered in lines 96,97,97 which is contradictory.

5. Line 87: FSH is a glycoprotein polypeptide hormone.

6. Sentences like the sentence in lines 93,94 and 95 need specific citations and generalization should be avoided.

7. The prevalence section should be more summarized and a conclusion should be added at the end of the section.

8. Title of 2.2.3 section should be revised. 

Author Response

(The authors gave the same response as above.)

Round 2

Reviewer 1 Report

 After reviewing the revised manuscript, I must say that the efforts to enhance clarity and accuracy have been successful. The improvements are noticeable and have significantly elevated the overall quality of the manuscript. The manuscript has undergone extensive editing and the result is a greatly improved version. I am satisfied with the changes and have no further comments.

Reviewer 2 Report

Authors revised their article completly and the article is now ready to be acepted.